

# Super Typhoons *Mangkhut* (2018) and *Saola* (2023) during landfall: comparison and insights for wind engineering practice

Yujie Liu[1], Yuncheng He[1,*], Pakwai Chan[2], Aiming Liu[3], Qijun Gao[4]

[1]Research Center for Wind Engineering and Engineering Vibration, Guangzhou University, Guangzhou, China
[2]Hong Kong Observatory, Hong Kong, China
[3]Shenzhen Meteorological Bureau, Shenzhen, China
[4]Department of Civil Engineering, Guangzhou University, Guangzhou, China

*Correspondence to*: Yuncheng He (yuncheng@gzhu.edu.cn, +86 13312861586)

**Abstract.** Offshore wind turbines are considerably sensitive to wind effects, and wind information of tropical cyclone (TC) lays the basis for their wind-resistant design and anti-TC operation especially in TC-prone areas. While the statistical characteristics of TCs have drawn continuous concerns, the specific features of some typical TC events, which are of practical importance for the daily operation of marine turbines, receive less attention in the wind engineering community. Super Typhoons *Mangkhut* and *Saola* are two of the strongest TCs impacting South China. Notably, although *Saola* was reported to

be more intense than *Mangkhut*, it resulted in much less severe impact and damage. This article presents a comparison study on these two TCs based on comprehensive usage of field records. Results suggest that both *Mangkhut* and *Saola* exhibited a concentric eyewall structure during development, but *Saola* completed the eyewall replacement before landfall, whilst *Mangkhut* failed to. Consequently, *Saola* evolved into a more intense and compact storm. By contrast, *Mangkhut* decayed consistently but still exerted extensive impact over a wider area. Consistent with these features, the wind characteristics of

*Mangkhut* and *Saola* also demonstrated noteworthy discrepancies. These findings provide useful insights for operation and maintenance strategies of coastal and offshore wind turbines.

**Keywords:** Wind Turbines, Tropical cyclone, concentric eye-wall, eye-wall replacement, RMW

## 1 Introduction

Wind energy, as a clean and renewable resource, holds a key place in the global strategies for energy transition and climate

change response (Martinez et al., 2023; Patel et al., 2022). With advancements in technology and reduced costs, wind power has become one of the most competitive electricity sources. As the demand for higher wind velocities and enhanced wind consistency grows, wind farm expansion has shifted towards more distant (Bento et al., 2019; Díaz et al., 2020).

In this trend, wind turbines also face increasingly stringent and complex natural environmental conditions (Qian et al., 2024). the impact of TCs is especially noteworthy (Matsui et al., 2002). In the open seas further from land, TCs encounter less

interference caused by terrain, leading to stronger winds that can result in aerodynamic failure in turbine blades (Dong et al.,



2018; Yang et al., 2020), distortion of tower structures, or trigger other catastrophic incidents such as the collapse for wind turbines (Sun et al., 2023). Concurrently, the massive waves and storm surges can inflict damage on the foundational structures of wind turbines, jeopardizing the normal operational integrity and power transmission at the wind farm site. If significant damage ensues, the post-TC repair work at the wind farm will be confronted with complicated working conditions and

exorbitant costs. In summary, the extreme wind velocities and wave-induced impacts caused by TCs present a rigorous test for the structural stability, durability, and safety of wind turbines. There conditions also impose heightened requirements on the TC-resilience capabilities of offshore wind farms during both their construction and operational maintenance phases (Kaldellis et al., 2016).

In the design of anti-TC phase for wind turbines, primary focus is placed on extreme wind velocities (Liu et al., 2019; Sheng

et al., 2021). researchers have extensively investigated the environmental loads and dynamic responses of turbines under extreme conditions, guided by design principles centered around extreme wind speeds (Gong et al., 2024; Li et al., 2019; Chen et al., 2020). These findings have significantly informed the design of TC-resistant turbines in coastal and offshore area. This approach involves initially estimating the TC-risk, typically defined by design wind speeds corresponding to various return periods, and subsequently utilizing this risk assessment to determine wind loads and wind-induced responses for the ultimate

limit states of the turbine (I.E.C., 2019). For such TC-risk assessments, a series of engineering models (e.g., TC wind field model and the filling model of TC central pressure) are employed usually in the scheme of Monte-Carlo sampling simulation to statistically quantify TC activities and associated extreme wind (Zeng et al., 2024). While these engineering models are able to depict the basic features of a common TC, they have not taken into account some complex but important TC characteristics (e.g., axial-asymmetry of TC pressure field) that can influence the simulation results markedly (He et al., 2020).

In the operational maintenance phase of coastal and offshore wind turbines, there should be a heightened focus on assessing the impact exerted by individual TC. Apparently, the aforementioned engineering models fail to cater for such requirements, as TCs often demonstrate complicated spatiotemporal evolution (Ren et al., 2022). It is true that the meteorological community has gained plentiful achievements on TCs' thermodynamic and kinetic structures as well as behind mechanisms, but there is still an evident gap between meteorology and wind engineering, and many insightful knowledge and wisdom in meteorology

has not attracted wind-engineers' attention timely, let alone being used for guiding wind-engineering practices (He et al., 2023). In summation, it becomes of great importance to focus on some typical TC events and explore their features so as to provide insights for the advancement and refinement of theories in wind engineering and practices in turbines technologies.

Super Typhoon *Mangkhut* in 2018 and Super Typhoon *Saola* in 2023 are the two strongest TC events that have attacked South China during the past years. During the landfall of the two special TCs, various meteorological instruments, including

meteorological satellites, weather radar and wind profilers, worked effectively and offered a valuable opportunity to explore these TCs from a comprehensive and global perspective. It is also noticeable that although *Saola* was reported to be more intense than *Mangkhut*, it resulted in much less severe wind-induced impact and damage.

To this end, this article presents a comparison study on *Mangkhut* and *Saola*, with a primary aim of providing in-situ evidence for better understanding the above discrepancy and drawing relevant insights for wind engineering. The rest of this article is



organized as follows. In Section 2, the two typhoons and main adopted devices and datasets are presented. Section 3 discusses the discrepancies of the two typhoons' intensity and influence area as well as the potential reasons, while Section 4 presents typical results of the TC wind. Main findings are summarized in Section 5.

## 2 Introduction of typhoon and dataset

### 2.1 Super Typhoons *Mangkhut* and *Saola*

Typhoon *Mangkhut* is the 22nd numbered TC event (No.: 1822) in the 2018 typhoon season over the Northwestern Pacific Basin (NPB). It originated around the middle part of NPB on 7th September, and developed into a Super Typhoon and reached its peak intensity prior to making landfall on Luzon on 14th, with the maximum sustained surface wind estimated at 70 m/s. *Mangkhut* made landfall near Zhuhai City of China as a strong typhoon on 16th and penetrated 780 km into mainland China before dissipating. During the typhoon's passage around Hong Kong (HK), Hong Kong Observatory (HKO) issued Hurricane

Signal No. 10 (which represents the highest warning level in this region and indicates that sustained surface winds of 33 m/s or greater have been observed) for 10 hours, which is the second longest since record-keeping began in 1949. *Mangkhut* caused extensive and severe wind and flood damage in China, affecting over 4 million people across six provinces. The storm resulted in the collapse or damage of more than 14,000 houses, with the direct economic losses exceeding 14 billion RMB (Li et al., 2022).

Typhoon *Saola* is the 9th numbered TC event (No.: 2309) in the 2023 TC season over NPB. It originated as a tropical depression on 24th August and gained iconic status as it surpassed *Mangkhut* in strength. On 30th, *Saola* passed over the Babuyan Islands in the northernmost part of the Philippines as super typhoon. At 22:00 on 1st September, the storm traversed Hong Kong with the central maximum winds reaching 58 m/s, and HKO issued the Hurricane Signal No. 10. Subsequently, *Saola* continued its westward movement along the coastal direction and eventually made landfall on the Leizhou Peninsula in Guangdong Province

of China. *Saola* mainly exerted its influence through heavy rains and severe floods. Reportedly, about 1.0 million individuals were evacuated in four provinces of China.

Figure 1 exhibits the evolution of TC track and intensity for *Mangkhut* and *Saola* when they moved close to HK, based on the best-track data issued by Chinese Meteorological Agency (CMA), HKO, Joint Typhoon Warning Center (JTWC) and Tokyo Climate Center (TCC). Note that the best-track data issued by different meteorological institutes may vary from one another

noticeably especially for TC intensity. Despite such discrepancies, it is observed that: (1) *Mangkhut* moved faster (with a translational speed varying in the range of 29-36 km/h) than *Saola* (11-16 km/h), (2) *Saola* got much closer to the Southeastern coastline of China than *Mangkhut* before making landfall, with the nearest distance between HK and the TC's center track being ~30 km for *Saola* and ~100 km for *Mangkhut*, (3) *Saola* maintained as a super typhoon when it got closest to HK and Shenzhen, which was more intense than the sever typhoon level for *Mangkhut*. Actually, from Figure 1(b), *Saola* was reported

to be distinctly stronger than *Mangkhut* during the landfall period.





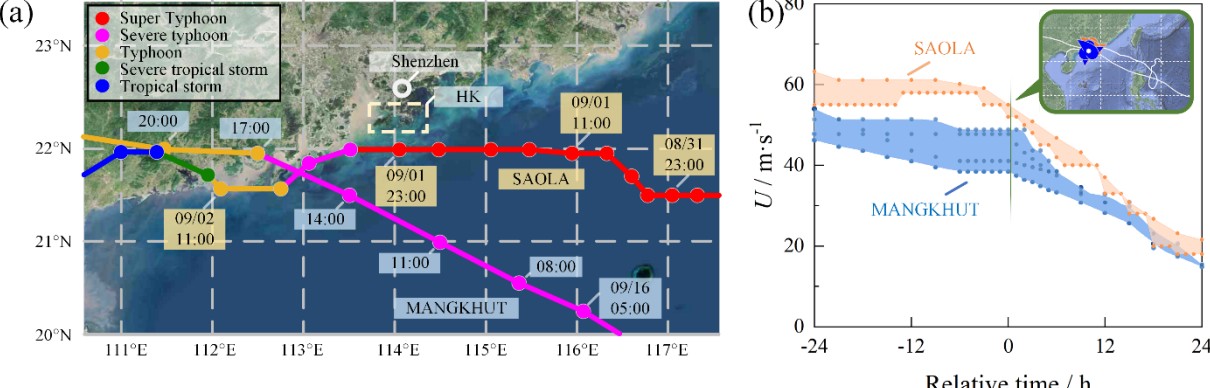

**Figure 1: Evolution of the track and intensity of *Mangkhut* and *Saola*. (a): center track from CMA, (b) TC intensity in form of envelopes based on the best-track data issued by CMA, HKO, TCC and JTWC. Relative time = 0 (from © Google Maps) in (b) denotes the moment when the storms moved closest to HK, i.e., at 14:00 on 16th Sep 2018 for *Mangkhut*, and at 22:00 on 1st Sep 2023 for**
***Saola* (hereafter)**

## 2.2 Devices and datasets

The data adopted in this study include the remote sensing observations via satellites and the field measurements obtained via various ground-based devices that dispersed in HK and Guangdong Province as shown in Figure 2. The infrared satellite

imagery is sourced from the CIMSS (Cooperative Institute for Meteorological Satellite Studies) database (https://tropic.ssec.wisc.edu/archive/index.php) which is procured by Himawari satellite series of Japan. These infrared satellite images offer a resolution between 1.0 and 2.0 km, providing temperature measurements across a wide spectrum from 180 to 310 Kelvin at 3-h intervals.

There are 86 national meteorological stations dispersed in the Guangdong Province of China, and each of them can provide

synchronous records of wind (in terms of horizontal speed and direction), ambient pressure, temperature and humidity at a near ground level that are updated in every 2 minutes. These records are mainly used to quantify the TC pressure field in this study.

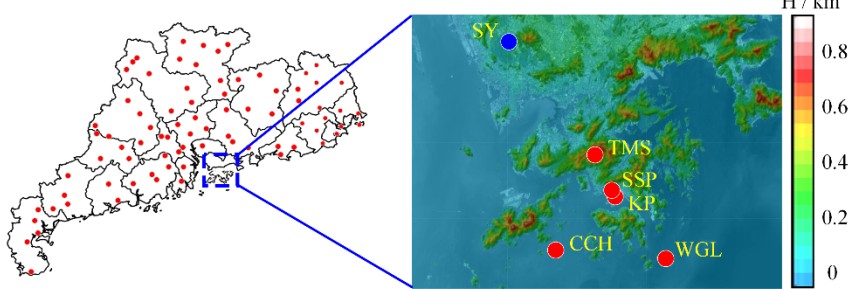

**Figure 2: Layout of metrological sites in HK and Guangdong Province: the red dots in the left picture correspond to the 86 natural**
**meteorological stations**



There are also dozens of weather stations established in HK and Shenzhen, and this study selects 6 of them (Figure 2) to analyze the thermodynamic features and wind characteristics of the two typhoons. Among these stations, SY sites around a suburb terrain in Shenzhen whose periphery is dominated by a long fetch of high-rise buildings, SSP and KP are located in the

Kowloon district of HK which is characterized by a built-up terrain, while TMS, CCH and WGL are respectively situated at the zenith of the Tai Mo Shan Mountain (955.2 m above local ground), Cheung Chau Island (71.9 m) and Waglan Island (55.8 m) in HK. All the above stations in HK except SSH can provide minute-to-minute updated records of wind (in terms of 1-min mean speed and direction as well as 3-sec peak gust within each 1 minute period), pressure and humidity at a near-ground level, while the two sites of SY and SSP are equipped with Doppler Radar wind profilers whose detection range can span from

315 m to 9223 m with 202.5 m gate intervals at SSP and from 100 m to 5790 m with 60 m (below 2560 m) or 120 m (above 2560 m) gate intervals at SY. The wind profiler records at SSP and SY are respectively updated in every 10 and 30 minutes. All the collected data were subjected to quality control (QC) procedures before being automatically recorded in accordance with the built-in modules. Meanwhile, additional QC measures were also conducted annually to further improve the credibility and quality of the datasets utilized for the following analysis, in a way consistent with those documented in our previous studies

(He et al., 2018). Detailed descriptions of the devices and QC techniques are referred to Reference (He et al., 2016).

Besides the above devices, this study also employs a high-performance Doppler weather radar system to provide ground-based radar echoes during the landfall periods of *Mangkhut* and *Saola* (http://www.nmc.cn/publish/radar/guang-dong/guang-zhou.htm). The radar system offers a coverage radius of approximately 460 km at about 10 km height, and update remote sensing records at 6-min intervals. It boasts a fundamental reflectivity detection range from 0 dBZ to 75 dBZ, coupled with an

image horizontal resolution of 1.0 km.

## 3 TC intensity and impact area

### 3.1 From main-circulation perspective

The main-circulation structure for a matured typhoon in the horizontal plane has long been acknowledged: a calm and cloudless TC eye normally in a circular or an elliptic shape sites around the center of the storm, which is surrounded by a ring of outward

tilting cloud wall (i.e., TC eyewall) where the strongest wind and heaviest rain exists; while rain-bands meander outside and continuously transport moisture as well as kinetic energy from peripheral regions to the inner core. The morphological characteristics of each of the TC's main-circulation components and their interactions may influence the intensity and impact area significantly. Typically, a stronger TC tends to possess a smaller eye/eye-wall (He et al., 2020).

Figure 3 shows infrared satellite snapshots taken at 4 moments during the landfall period for *Mangkhut* and *Saola*. The

grayscale values in the image represent the temperature radiation in the detection area, with darker shades indicating higher temperatures and white denoting the surface with the lowest temperature. As clouds at higher altitudes in the troposphere generally exhibit colder temperatures, lower gray values (i.e., brighter clouds) in the images can be interpreted as an indicator



of higher-altitude clouds with more intense convection. Note that as the top of a matured TC is usually capped by a shield of dense cloud that is formed by the ice crystals out-flowing from the eyewall, one may not be able to observe detailed main-circulation structure from the infrared satellite image.

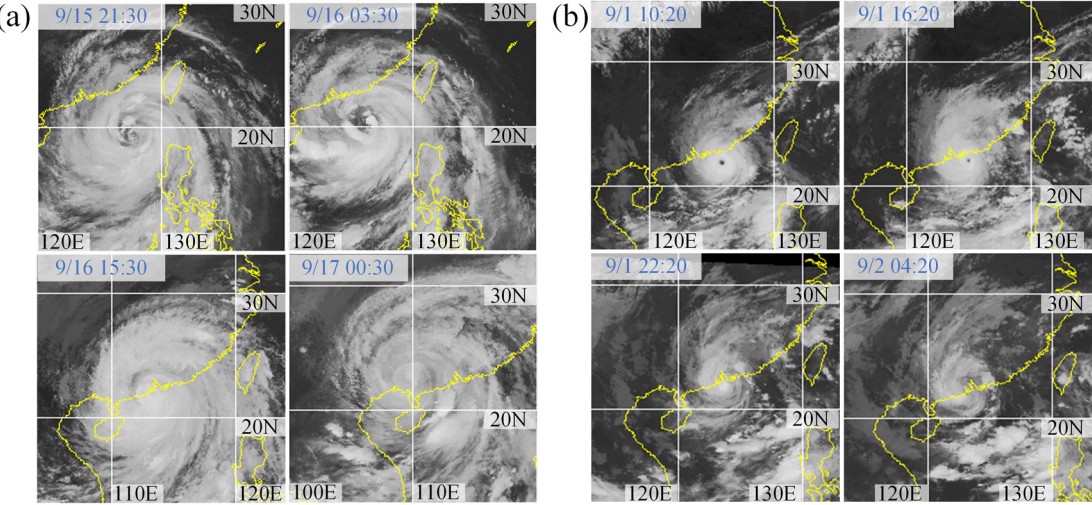

**Figure 3: Infrared satellite snapshots of *Mangkhut* (a) and *Saola* (b) at 4 selected moments during landfall. Each photo spans an area of 20°×20°, or about 2,200×2,200 km (from CIMSS 2018)**

Comparing Figure 3(a) and Figure 3(b) reveals that: (1) the cloud system of *Mangkhut* was distinctly larger (with a diameter of 15°) than that of *Saola* (5°), which is consistent with the much wider impact area for *Mangkhut*; (b) there was an evident typhoon eye (with a diameter of 25 km) surrounded by a ring of convective tower (corresponding to the eyewall) for *Saola* before landfall, which indicates the storm's high-level intensity. By contrast, no such typical TC eye and convective tower were observed for *Saola*. Instead, the central area before landfall was occupied by an irregular region of loose or flimsy cloud which carried some characteristics of TC eye.

Based on the infrared satellite images, one can estimate TC intensity typically via the Dvorak technique (Dvorak, 1984; Olander et al., 2019). Recently, the authors have used deep learning (DL) technology to automatically quantify TC intensity and locate the storm center based on satellite images (Tong et al., 2022; Long et al., 2022). Figure 4 presents the DL aided estimations of TC intensity for *Mangkhut* and *Saola* during the landfall period. The obtained results are similar to those presented in Figure 1(b), both of which demonstrate *Saola* was more intense than *Mangkhut*.



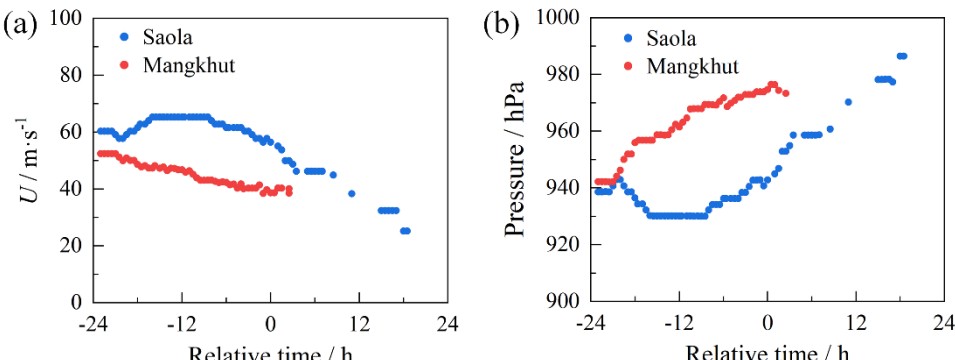

**Figure 4: Intensity of *Mangkhut* (a) and *Saola* (b) estimated via DL technique based on satellite images in terms of both central pressure and maximum surface sustained wind speed**

To detail the main-circulation (in particular the rain-band) structure of the two typhoons beneath the cloud shield, Figure 5 exhibits the echo-grams from the ground-based weather radar. The reflectance values in each gram indicate the density of cloud/precipitation, with warmer colors corresponding to the stratified rainband complex or the eyewall. As demonstrated, the echo-grams of *Mangkhut* were dominated by large-reflectance-featured stratified rainband complex which span an area of ~800 km in diameter, and the maximum reflectance values existed dispersedly in such rainband complex. By contrast, the rainbands of *Saola* were much smaller (~400 km) whose periphery was dominated by small-reflectance-featured convective cloud cells, while the maximum reflectance values only appeared in the inner core (corresponding to the eyewall). These findings again suggest *Saola* was an intense and compact typhoon, while *Mangkhut* exerted severe impact in a much wider scope.

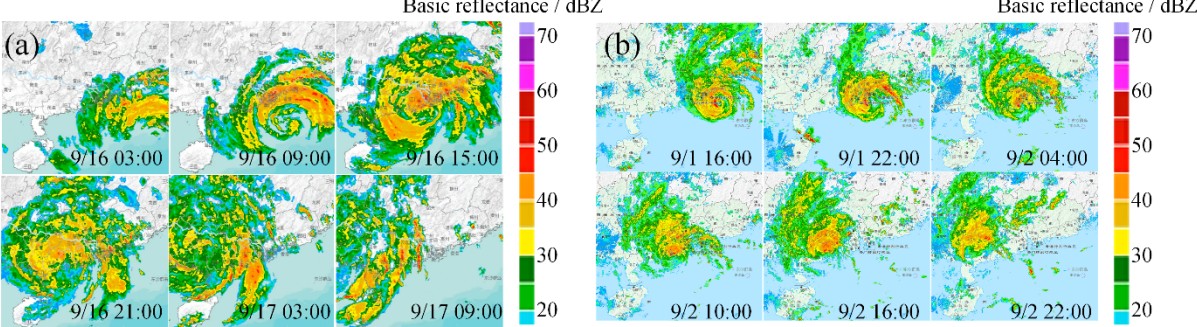

**Figure 5: Echo-grams from ground-based weather radar for *Mangkhut* (a) and *Saola* (b). All grams correspond to a geographical area of 1000×1000 km (from 108° E/18° N to 119° E/27° N)**





### 3.2 From secondary-circulation perspective

The secondary-circulation of TCs can be briefly described as an in-up-out-down overturning movement of flows in the vertical

plane (Wu et al., 2021). The inflow layer occupies the lowest several kilometers where atmospheres gain moisture and energy from sea surface as they flow inward. When the inflows reach to the eyewall, they change to rise and meanwhile release latent heat so that the inner core can sustain as a hot tower. When the rising flows approach to TC's top, they begin to go outward. The out-flowing atmospheres then turn to descend as they reach to the storm's peripheral areas, making the weather at the near surface to be distinctly hot and dry.

Figure 6 depicts the radial distribution of dry-bulb temperature ($T_{air}$) and relative humidity ($RH$) recorded at three meteorological stations in HK (refer to Figure 2) during the landfall of *Mangkhut* and *Saola*, where Negative (positive) values of radial distance denote the storm was approaching to (moving far away from) the study sites, hereafter. Note that as the two TCs had not directly passed through HK, there is an absence of records at TCs' most inner region.

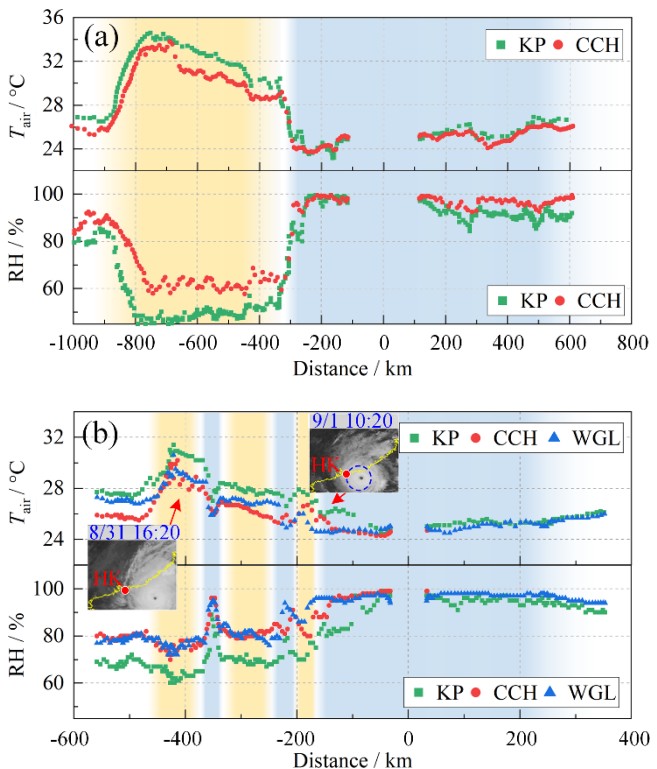

**Figure 6: Radial distribution of dry-bulb temperature (Tair) and relative humidity (RH) at 3 sites in HK during the landfall of**
***Mangkhut* (a) and *Saola* (b)**

From Figure 6(a), when *Mangkhut* was 900 km away from the study sites, the measured $T_{air}$ increased whilst $RH$ dropped

sharply, reflecting that the subsidence flows from the storm's periphery started to influence HK. The hot and dry subsidence flows became most evident when the radial distance was about 750 km, with the maximum change of $T_{air}$ and $RH$ equal to 8°C





and -30%; while they got minimized when the distance was 300 km, after which the study sites were influenced by stratified rainbands and the atmosphere turned to be saturated. Thus, the subsidence flows spanned a radial distance of ~600 km before *Mangkhut*'s landfall. However, no typical evidence of such subsidence flows was observed after the storm's landfall.

Results for *Saola* demonstrate some similar features, including the evident increase of $T_{air}$ and noticeable decrease of $RH$ when the storm's periphery moved close to HK before landfall. The TC atmosphere then tended to be saturated as the study sites got close to the storm's inner core. The typical difference between the two typhoons lies in that the observed subsidence effects were much weaker for *Saola*, not only from the perspective of the much less maximum change of $T_{air}$ and $RH$ but also from the aspect of considerably limited span of radial distance associated with such subsidence flows. Meanwhile, there were also

two episodes before *Saola*'s landfall (with the central radial distances equal to 350 km and 225 km) in which $T_{air}$ decreased and $RH$ increased abnormally. These two episodes are expected to be caused by the spiral rainbands that impacted the study sites intermittently. In sum, Figure 6 provides further evidence for the much larger impact area of *Mangkhut* than that of *Saola*.

**3.3 Quantification of wind-impact area**

The impact area of TC wind can be quantified in terms of the radius of maximum wind (RMW), which can be determined via

the radial profile model of TC pressure, e.g., proposed by (Holland, 1980):

$$P(r) = P_0 + \Delta P_0 \exp[-(RMW / r)^B] \tag{1}$$

where $P(r)$ denotes the mean-sea-level pressure at radial distance $r$, $P_0$ denotes the pressure at the TC center, ambient pressure; $\Delta P_0$ is the difference between the ambient pressure and $P_0$ (or the central pressure deficit), $B$ is a non-dimensional coefficient which governs the shape of the radial profile.

Figure 7 shows the modelling results of the pressure field for *Mangkhut* and *Saola* via the above profile model, based on the therm-hydro-baro measurements at the 86 national meteorological stations distributed in Guangdong (Figure 2). From Figure 7(a), the pressure field of *Saola* varied significantly for $r$<50 km, beyond which the radial profile tended to level off, reflecting that large pressure gradient existed only within a limited inner region of the storm; while for *Mangkhut*, the pressure varied gradually in a much larger radial range. The above observations suggest that strong winds only existed within a limited inner

region of *Saola*, but they could remain in a much wider radial range of *Mangkhut*. Figure 7(b) shows the time histories of *RMW* and $B$ around the storms' landfall. As expected, the values of *RMW* for *Mangkhut* (on the order of 100 km) were considerably larger than those for *Saola* (on the order of 10 km), so were the conditions for $B$. These findings provide further evidence to support *Mangkhut* exerted wind-impact in a much wider scope.



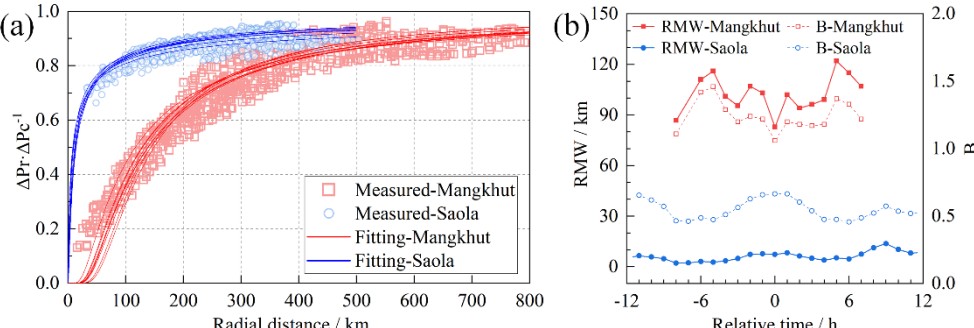

**Figure 7: Modeling results of TC pressure field for *Mangkhut* and *Saola*: (a) normalized radial profiles, (b) RMW and B (0-h marks the time when the storm just got landfall)**

From the above discussions, it is of great importance to quantify the wind-impact area for anti-TC practices. Unfortunately, such information has not been covered in the TC best track data (which only include the TC center and TC intensity in terms

of central pressure and maximum sustained surface wind). Although RMW has been utilized in wind engineering for assessing TC hazards, the database of RMW is badly lacking. The values of RMW for TCs over many regions can only be estimated roughly via some empirical formulas that have been established on the basis of TCs over some studied regions. Meanwhile, the RMW is conventionally regarded as exhibiting a negative correlation with central pressure, signifying that the more intense a TC becomes, the smaller the RMW tends to be the finding consistent with the observations derived from this study.

**3.4 Potential reasons for observed discrepancies**

In this study, the discrepancies between the two typhoons in terms of intensity and impact area are primarily attributed to the different evolution of eyewall during their development. It has been well acknowledged that for intense TCs over deep seawater, an outer eyewall may form outside the initial (or inner) eyewall, and the storm demonstrates a concentric eyewall structure. As illustrated in Figure 8, between the two rings of eyewall, there exists the so-called moat which plays a role similar to the

TC eye. The presence of moat and outer eyewall prevents the influx of water vapor and energy in outer regions from being transferred into the inner eyewall, making the inner eyewall to decay. Meanwhile, the outer eyewall tends to shrink and gradually replace the inner eyewall. The above process is termed as the eyewall replacement (ER) (Houze et al., 2007). Previous studies have shown that the ER process can affect the TC intensity significantly. Usually, the storm first decays, then recovers and finally becomes even stronger than the initial status.

Figure 9 presents the ER process of *Mangkhut* and *Saola*. From Figure 9(a), *Mangkhut* exhibited an evident con-centric eyewall structure around 21:30 on 14 Sep. when it was to get landfall on Luzon in Philippine. However, the storm failed to finish the ER cycle, as: (a) it first made landfall on Luzon, which destroyed its inner structure, and (b) after the first landfall, *Saola* moved to the South China Sea and approached to the southeast coastline of China where the ambient conditions became unfavorable for its further development. Consequently, the inner eyewall tended to dissipate but the outer eyewall stopping



shrinking and finally disintegrated into a pair of rainband complexes. In accordance with the incomplete ER, the maximum TC wind firstly existed around the inner eyewall and then evolved to appear in a much wider range of the disintegrated outer eyewall. Overall, *Mangkhut* was weakened but it exerted wind impact in a much wider scope after such an incomplete ER process.

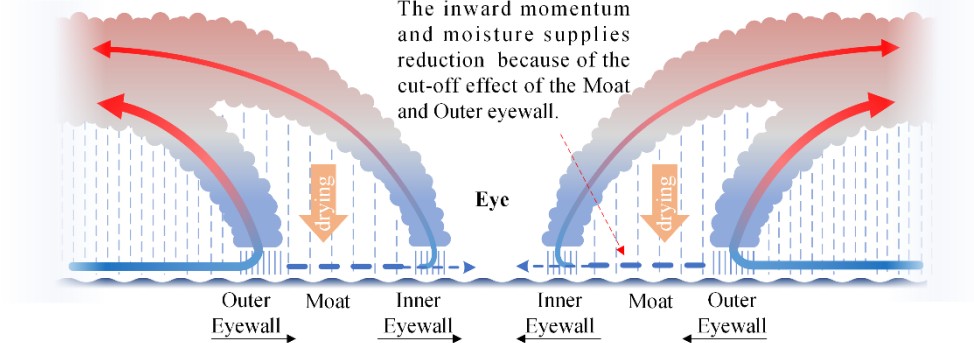

**Figure 8: Schematic diagram of concentric eyewall structure and ER. The density of rain lines corresponds to the relative rainfall. Thick arrows indicate the potential flow of water vapor and energy being carried, while thinning dashed arrows signify that the flow has been obstructed and weakened. Black arrows indicate inner and outer eyewalls progressing towards the TC center during the ER cycle. To provide clearer details, the near-surface portion of the eyewall has been suitably enlarged (Houze et al., 2007)**

For *Saola*, the detection results from Synthetic Aperture Radar (SAR) reveal a complete ER cycle before the storm's landfall. As demonstrated in Figure 9(b), there was a single eyewall at 17:44 on 28 Aug. 2023 at a distance of ~10 km away from the TC center, and the maximum wind was estimated to be 60 km/s. On 29 Aug., the concentric eyewall structure appeared, and there were two peaks in the radial profiles of TC wind at 10 km and 40 km, which respectively corresponded to the inner eyewall and outer eyewall. The maximum wind was estimated to be 43 m/s, which is markedly smaller than that on 28 Aug. The ER cycle ended on 30 Aug., when there was only one peak in the radial wind profile at a distance less than 20 km which corresponded to the outer eyewall that had already replaced the initial inner eyewall. The maximum TC wind was estimated to be 64 m/s, which is a bit stronger than that (60 m/s) at the beginning of the ER process. Overall, the successful completion of ER process enhanced *Saola*'s intensity.





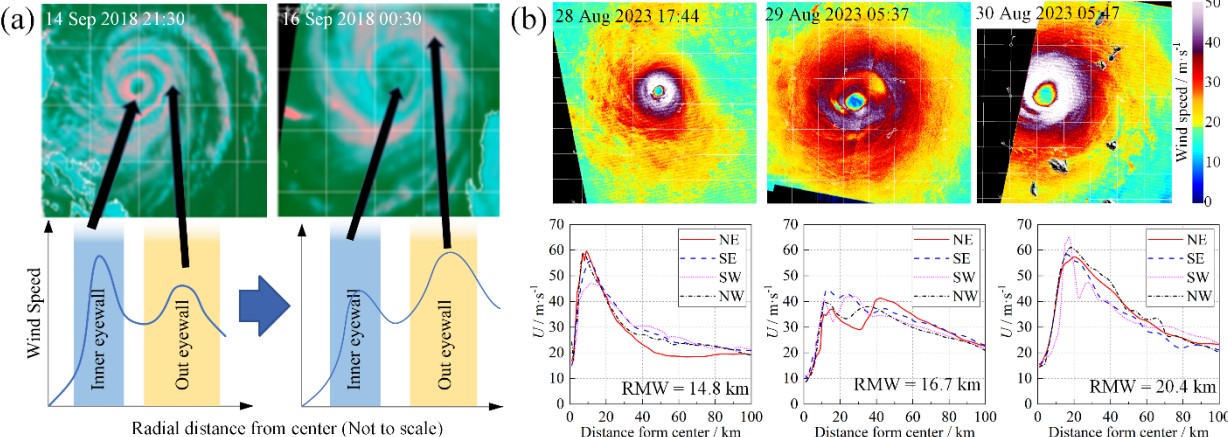

**Figure 9: ER cycle for *Mangkhut* (a) and *Saola* (from** https://www.weather.gov.hk/informtc/ *Mangkhut*18/windstructure.htm) **(b).**
**Cloud image represents radial wind profiles obtained through the conversion of cross-polarized normalized radar cross section**
**backscatter (from** https://www.star. nesdis.noaa.gov/socd/ mecb/sar)

The varied status after the aforementioned ER processes for the two typhoons can be also seen from the satellite images shown
in Figure 3. For *Saola*, the compact TC eye existed almost until the storm made landfall. However, for *Mangkhut*, the central
area before landfall was occupied by an irregular but much huger region of loose or flimsy cloud which should correspond to
the eye and decayed inner eyewall.

There are also some other potential reasons for the discrepancies of the two typhoons. One may be correlated to the TC track.
As illustrated in Figure 1(a), the track of *Saola* was much closer to the coastline of China before it got landfall than that of
*Mangkhut*. As the drag effect was distinctly larger over land than over seawater, *Saola* tended to be more severely weakened
for the parts outside of the inner core. Meanwhile, as discussed previously, when *Mangkhut* passed through Philippine, the
inner structure was disintegrated in a much wide region. By contrast, *Saola* had not been influenced by such effects, which
facilitated the storm to keep compact. Another influence factor is about the TC translational speed, which has been
demonstrated to be able to influence the TC intensity noticeably. An appropriate translational speed favors a TC to absorb
energy and water vapor from sea surface, whilst either a too large or a too small translational speed goes against it. As
introduced previously, the translational speeds of the two typhoons before landfall varied in a range of 29-36 km/h (*Mangkhut*)
and 11-16 km/h (*Saola*). The one for *Mangkhut* was a bit too large for the storm's recovery in strength.

## 4 Wind field

### 4.1 Mean vertical profile

The vertical profiles of TC wind during the landfall of *Mangkhut* and *Saola* are examined, based on the remote sensing
detection from the Doppler Radar Wind Profilers (DRWPs) at SSP (in HK) and SY (in Shenzhen). Figures 10-11 respectively

present the evolution of 30-min mean horizontal wind speed $U$ (Figure 10) and direction $\theta$ (Figure 11) with respect to the radial distance ($r$) between the study site and the con-temporal TC center. The radial distances of ±115, ±141, ±40 and ±65 km in figures respectively denote the closest distances between the TC tracks and the study sites. Figure 12 depict the results of

nominal vertical speed $W$ and associated signal-to-noise ratio (SNR-$W$). Note that the values of $W$ from a DRWP under rainy conditions should be interpreted as the falling speed (or roughly the intensity) of raindrops. It is also noted that SNR-$W$ can be utilized to determine the depth of mixing layer (which serves as a characteristic length of the depth of atmospheric boundary layer) and the location of melting layer (He et al., 2020).

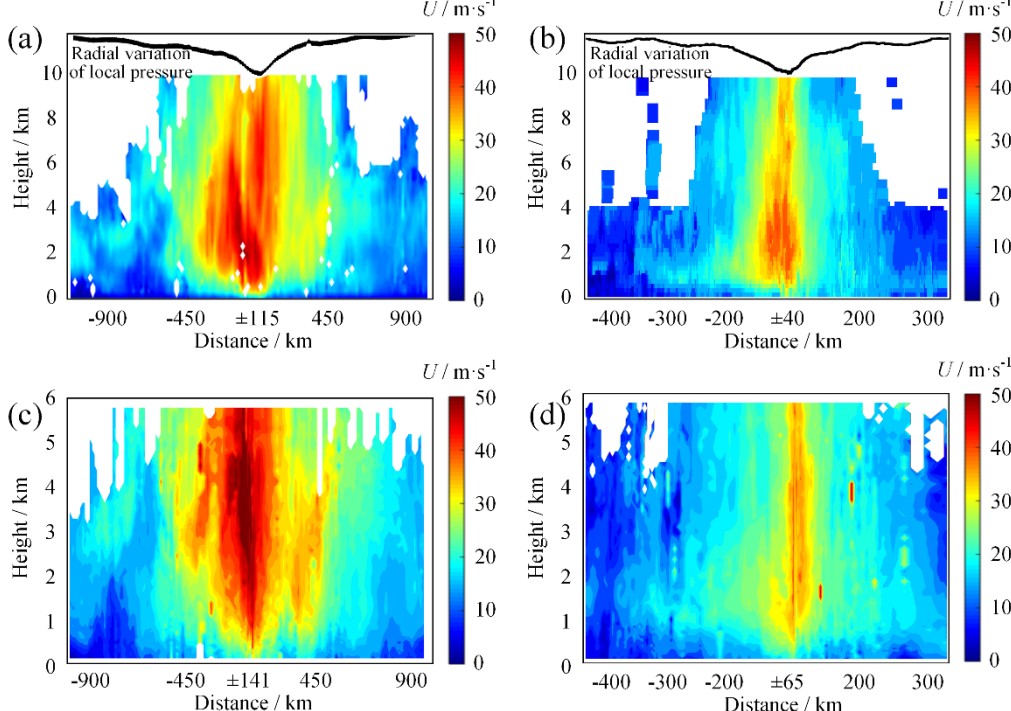

**Figure 10: Vertical profiles of 30-min mean horizontal wind speed in function of radial distance—(a):** *Mangkhut* **at SSP, (b)** *Saola* **at SSP, (c)** *Mangkhut* **at SY, (d)** *Saola* **at SY**

Results shown in Figure 11 suggest the existence of the melting layer at ~5 km. The precipitations below this layer were dominated by fast-falling raindrops, whilst those above it were dominated by slowly-falling ice crystals. The radial scope with

respect to the TC center at the study site below 5 km can be basically divided into 3 portions according to $W$: the inner portion dominated by torrential rains (-280<$r$<180 km for *Mangkhut* and -160<$r$<80 km for *Saola*), the outer portion with light or no rains (outside -500<$r$<470 km for *Mangkhut* and outside -250<$r$<250 km for *Saola*; corresponding to the TC periphery that was governed by convective cloud cells), and the middle portion governed by intermittent strong-to-moderate rainfalls. Each of the portions for *Mangkhut* was considerably larger than that for *Saola*, which reflects the much wider impact area of

*Mangkhut*. It can be also seen from Figure 12(c, d) that the depth of mixing layer in the outer portion for *Mangkhut* was on the





order of 3 km, compared to that of 2 km for *Saola*. The larger mixing layer depth of *Mangkhut* can be reasonably explained by the stronger secondary-circulation especially in terms of subsidence flows for this typhoon as discussed in Figure 6.

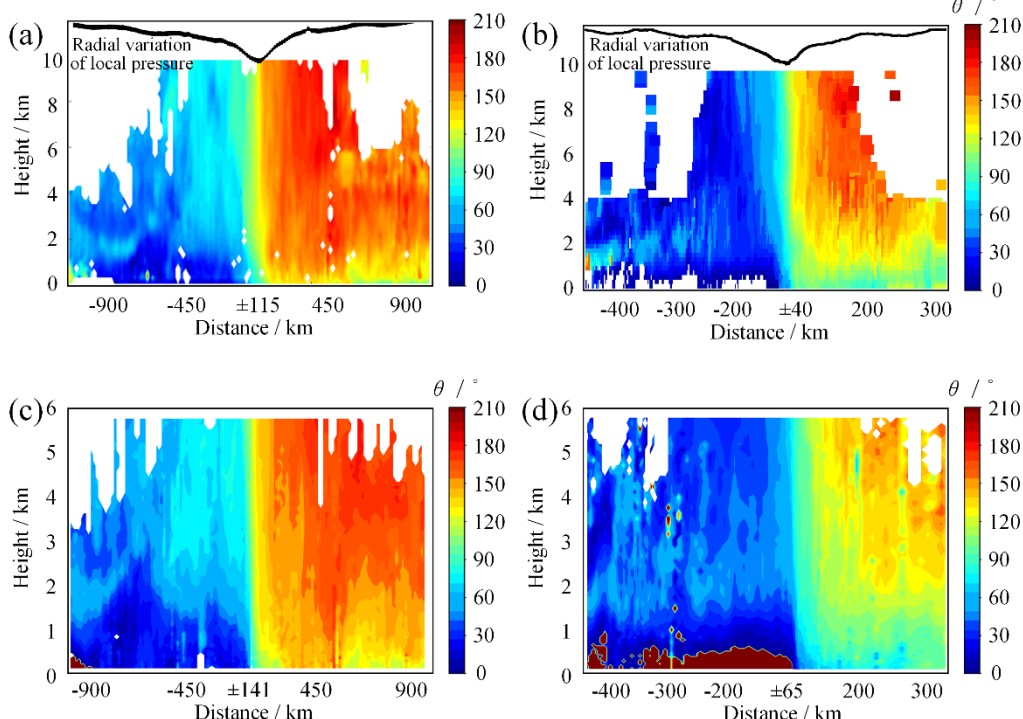

**Figure 11: The same to Figure 12, but for wind direction (*θ*)—(a):** ***Mangkhut*** **at SSP, (b)** ***Saola*** **at SSP, (c)** ***Mangkhut*** **at SY, (d)** ***Saola***
**at SY.**

Scrutinizing Figures 10(a, b) and 11(a, b) in conjunction with Figure 12 reveals that the inner portion contained the strongest wind, and wind direction within this portion changed significantly (exceeding 100°), with the wind before and after the TC passage respectively dominated by north and southeast flows. By contrast, the wind in the outer portion was quite weak,
although the one for *Mangkhut* was relatively larger than that for *Saola*, owing to its more remarkable subsidence flows. Overall, as *Saola* was more compact and intense than *Mangkhut*, the TC wind during the passage of *Saola* changed more sharply with respect to radial distance.



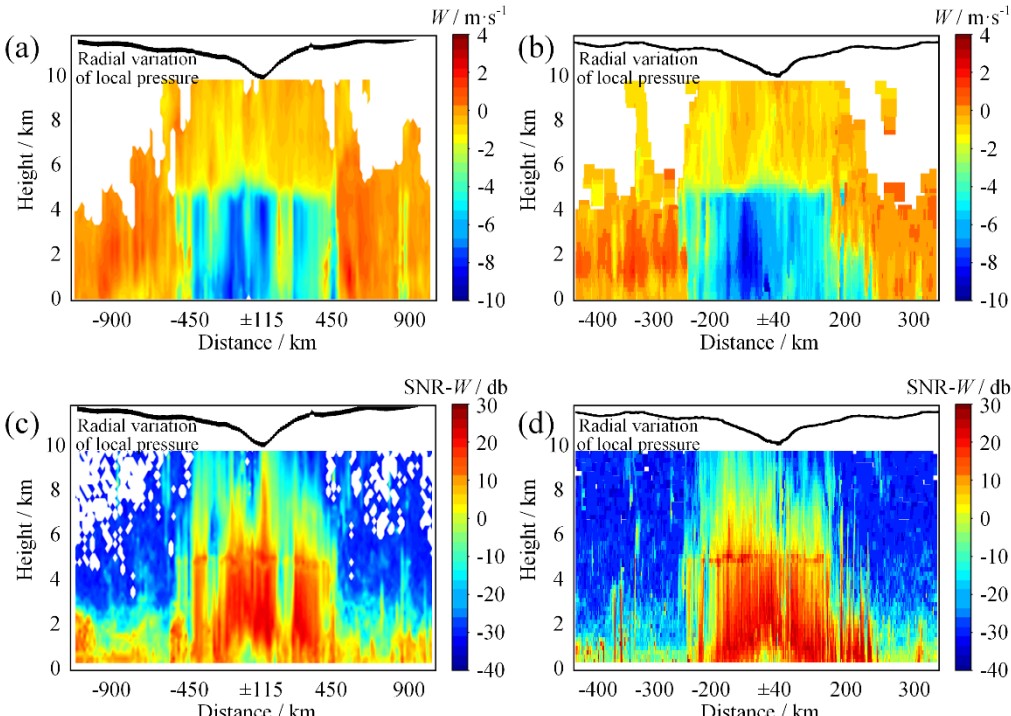

**Figure 12: Vertical profiles of nominal vertical speed (*W*) and associated signal-to-noise ratio (*SNR*) at SSP—(a): *W* for *Mangkhut*,**
**(b): *W* for *Saola*, (c): *SNR* for *Mangkhut*, (d): *SNR* for *Saola*.**

It is interesting to find from Figure 10 that the TC wind in the inner portion for *Mangkhut* at SY was stronger than that at SSP, whilst the one for *Saola* at SY turned to be smaller than that at SSP, despite the fact that SY was located consistently farther away from both typhoons (about 25 km). As discussed previously, the incomplete ER process resulted in a much larger RMW 335 for *Mangkhut*. It is speculated that although SSP was situated closer to the TC center, SY should be nearer to the maximum wind of the storm. However, as *Saola* finished the ER cycle, it became considerably compact, and both stations were well located outside of *Saola*'s maximum wind. Since SY was farther away from the TC center, the wind there should be weaker. To detail the TC wind field, Figure 13 presents the vertical profiles of 2-h mean horizontal wind speed measured by the DRWPs. Figures 13(a, b) correspond to the results at SSP during the passage of *Mangkhut* and *Saola*, respectively. Figures 13(c, d) 340 compare the measured profiles against the stipulations in the national wind load code of China (GB 50009-2012, 2012) associated with 4 categories of terrain conditions. For both typhoons, the vertical profiles demonstrate a low-level-jet (LLJ) structure, which differs significantly from the traditional depiction of wind profile that is featured by unchanged wind above the gradient height. Meanwhile, the LLJ height where the maximum wind appears evolved evidently during the landfall period. The value is found to vary from 1.5 km to over 3 km, which is distinctly larger than the gradient height recommended in the 345 wind load code (300-550 m according to terrain category). From Figure 13(c, d), remarkable difference exists between the





measured profiles and code stipulations. Apparently, using such wind codes for anti-TC design or resilience assessment of buildings (especially of high-rise buildings) may result in significant estimation errors.

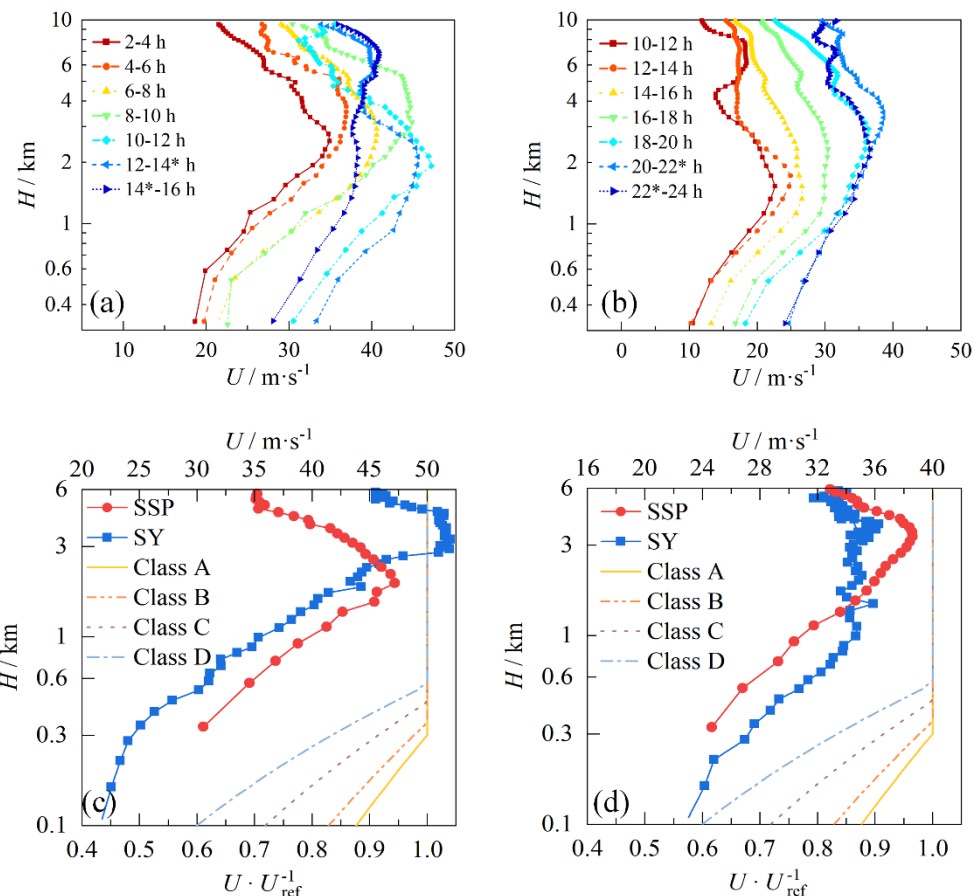

**Figure 13: Vertical profiles of 2-h mean wind speed—(a): results at SSP for Mangkhut, (b) results at SSP for *Saola*, (c): profiles containing the maximum wind at SSP and SY during Mangkhut compared to stipulations in wind load code of China, (d): similar to (c) but for *Saola*. 14\* and 22\* in (a, b) indicate the landfall moments of Mangkhut (14:00) and *Saola* (22:00).**

## 4.2 Gradient wind

Gradient wind is very important for analyzing boundary layer wind. In practice, one can first calculate the gradient wind field as it is more simple and deterministic, and then deduce boundary layer wind based on the results of gradient wind (Meng et al., 1997). Thus, the estimation accuracy of TC gradient wind directly affects the computational accuracy of TC boundary layer wind.

Gradient wind is also frequently used for converting between boundary layer winds with varied terrain categories, where the gradient wind above different terrain setups is usually assumed to be unchanged. Based on the assumption of unchanged



gradient wind, He et al. (2014) developed a standardization method to convert the measured surface wind into their counterparts under a reference condition (i.e., mean wind speed at 10 m above the terrain with roughness length $z_0$=0.03 m). The validity of the standardization method has been verified under strong monsoon conditions, but its effectiveness in TC situations remains to be further examined.

Given the pressure field as quantified by Equation (1), the TC gradient wind can be computed as follows:

$$U\left(\alpha, r\right) = \frac{1}{2}\left(U_{\mathrm{T}}\sin(\alpha) - fr\right) + \sqrt{\frac{1}{4}\left(U_{\mathrm{T}}\sin(\alpha) - fr\right)^2 + \frac{B\Delta P_0}{\rho}\left(\frac{RMW}{r}\right)^B \exp\left[-\left(\frac{RMW}{r}\right)^B\right]}$$


(2)

where, $U\left(\alpha, r\right)$ signifies the gradient wind speed with respect to the TC direction offset angle $\alpha$ and the radial distance $r$, $U_{\mathrm{T}}$ denotes the TC translational speed, $f$ represents the Coriolis coefficient ( $f = 2\Omega\sin(\phi)$, $\Omega=\pi/(12 \times 3600)$ symbolizes the Earth angular velocity of rotation, $\varphi$ denotes latitude) and $\rho$ indicates air density.

The calculated results via the above model are compared against the measured gradient winds which are recognized as the
maximum $U$ below 3 km in each mean vertical profile from the DRWPs, as shown in Figure 14(a). The measured gradient winds are also compared with their counterparts obtained via the standardization method developed by He et al. (2014), as shown in Figure 14(b). Meritorious attention is warranted to the observation that, owing to the influence of supergradient winds, the measured wind velocities frequently surpass the true gradient winds (Kepert, 2001a). The investigative outcomes presented by Kepert (2001b) suggest that this disparity might potentially exceed a magnitude of ten percent.

As illustrated in Figure 14(a), the modeled results agree better with the measured gradient winds for *Saola* than for *Mangkhut*. Note that super-gradient wind exists widely in the eyewall and rainband regions of TCs, which is regarded as a main reason for the appearance of LLJ in TC vertical wind profile (Kepert, 2010). This phenomenon partially accounts for the discrepancies between the modeled results and the measurements in Figure 14(a), since Equation (2) has not taken into account the vertical wind component (*W*) which plays an essential role in the generation of TC super-gradient wind (Kepert, 2010). However, the
measured results for *Mangkhut* are too large compared to the modeled values, and considerable errors should exist in the modeled gradient wind. On the one hand, the effects of con-centric eyewall or ER may not be neglected for this typhoon during the study period, but Equation (1) does not account for such effects. On the other hand, as highlighted in the investigative outcomes presented by He et al. (2021), the pressure field of *Mangkhut* demonstrated asymmetric features when the storm came close to the coastline of China, which cannot be depicted by Equation (1). Overall, the modeling errors are largely
attributed to the inaccurate reproduction of the TC pressure field by using Equation (1), except for the super-gradient effect.





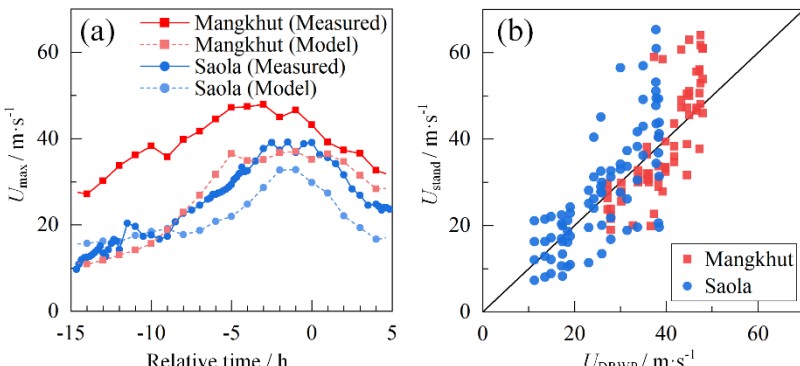

**Figure 14: TC gradient wind measured by DRWPs compared with modeling results (a) and those deduced via the standardization method (b).**

From Figure 14(b), the deduced results of gradient wind via the standardization method basically agree with the measured gradient winds for wind below ~40 m/s, which reflects the applicability of the assumption of unchanged gradient wind (or the developed standardization method) in such situations. However, under ever stronger wind conditions, the two kinds of results start to differ markedly, suggesting the assumption of unchanged gradient wind and therefore the standardization method becomes inappropriate. Actually, the standardized gradient wind here is calculated through multiplying the measured surface

wind by a corresponding correction factor (i.e., $K_2$ as reported in reference (He et al., 2014)), but the standardization results (including the values of $K_2$) presented in He et al. (2014) were computed on the basis of field measurements collected during strong monsoons. As discussed previously, TC winds differ from monsoon winds especially in TC's inner region where the LLJ feature as well as the gradient of TC wind field along radial distance become evident. Thus, one should take great cares when assuming unchanged gradient wind and converting between TC surface winds with varied terrain setups in a way similar

to the one for monsoons.

**5 Concluding remarks**

Based on field measurements collected from a variety of devices, this study compares the structural features and wind characteristics of two intense typhoons, with a highlight focused on the intuitively abnormal correlations between the TC intensity and associated impact area. Potential reasons for the abnormal correlations have been presented, and some useful

insights for engineering practices of FOWT have been drawn. Main findings and conclusions are summarized as below.

(1) *Saola* was more intense but compact than *Mangkhut*. During the landfall period, the maximum TC strength of *Saola* exceeded 60 m/s, compared to that of ~50 m/s for *Mangkhut*. However, *Mangkhut* exerted its impact in a much larger scope from both the main-circulation and secondary-circulation perspectives. Specifically, *Mangkhut* had a huge cloud system whose diameter exceeded 1500 km and the stratified rainband complex spanned ~800 km. By contrast, the diameter of *Saola*'s cloud

system was about 500 km, and the rainbands occupied an area of ~400 km whose primary portion was constrained within an



even narrower inner region. Meanwhile, *Mangkhut* resulted in considerably stronger (in terms of radial span and the maximum change of $T_{air}$ and $RH$) subsidence flows at the study site than *Saola*, which again reveals its intense impact in a distinctly wider area. Results of *RMW* provide further evidence to quantitatively compare the wind-impact areas for these two typhoons. It was found that the *RMW* of *Mangkhut* exceeded 100 km, which was one order of magnitude larger than that of *Saola*.

(2) The aforementioned discrepancies were primarily attributed to the varied eyewall replacement (ER) processes of the two typhoons. Both *Mangkhut* and *Saola* demonstrated a concentric eyewall structure during the development stage. Once the outer eyewall formed, the inner eyewall tended to decay owing to the cutting off of energy and water vapor transferred from TCs' peripheral areas. For *Saola*, as the inner eyewall decayed, the outer eyewall shrunk and gradually replaced the inner eyewall. During this process, the TC strength first decreased and then recovered and finally became a bit stronger than the initial status.

By contrast, *Mangkhut* failed to finish the ER cycle. The outer eyewall did not shrink; instead it disintegrated into a pair of rainband complexes in a considerable large region. When the storms approached the study sites, *Saola* turned to be more intense and compact, whilst *Mangkhut* became weakened and exerted impact in a much wider scope. Other potential reasons were also presented, including the discrepancies of TC track.

(3) In accordance with the TC structural features, the wind characteristics of *Mangkhut* and *Saola* at the study sites also 425 demonstrated some noteworthy discrepancies. The radial scope with respect to the TC center below the melting layer can be basically divided into 3 portions: the inner portion dominated by torrential rain and strongest wind, the outer portion with light or no rains as well as weakest wind, and the middle portion governed by intermittent moderate-to-strong rainfall and wind. Each of the portions for *Mangkhut* was considerably larger than that for *Saola*. In the outer portion, due to the more intense subsidence flows, the mixing layer depth for *Mangkhut* was evidently larger (3 km) than that (2 km) for *Saola*. In the inner 430 portion, the wind of *Mangkhut* was stronger at SY than at SSP, whilst *Saola*'s wind turned to be smaller at SY than at SSP, despite the fact that SY was located consistently farther away from both TCs' tracks. Results of TC gradient wind suggest that there were noticeable modeling errors for *Mangkhut*, possibly due to the combined effects of super-gradient wind and inaccurate reproduction of the TC wind field. Meanwhile, as the gradient wind changed rapidly with radial distance around TC inner region, the assumption of unchanged gradient wind may not be exploited appropriately, e.g., for converting between 435 boundary layer winds with varied terrain setups.

In sum, TCs possess complex structural features and wind characteristics which may evolve spatiotemporally and can be hardly depicted by most widely utilized engineering models. As the scale increases of coastal and offshore wind turbines, there is a need for wind engineering community to focus more on individual TCs especially during the phase when they are situated within the nearshore zone. The observations presented in this study suggest the importance of quantifying both TC's strength 440 and impacting scope for assessing TC hazards. Meanwhile, detailed evolving features of TCs, e.g., concentric eyewall and ER process, should be also taken into account for some special TCs so as to better understanding their wind characteristics and wind-induced impact.



**Conflict of Interest**

No conflict of interest exits in the submission of this manuscript, and manuscript is approved by all authors for publication. I would like to declare on behalf of my co-authors that the work described was original research that has not been published previously, and not under consideration for publication elsewhere, in whole or in part. All the authors listed have approved the manuscript that is enclosed.

**Copyright Notice**

All maps, images, and aerial photography included in this manuscript are either owned by the author or used with explicit permission from the respective copyright holders.

**Code/Data Availability**

All publicly available data used in this study have been clearly cited and referenced within the manuscript. For any additional data or code that are not publicly accessible, please contact the corresponding author for further information.

**Author contribution**

**Yujie Liu**: Formal analysis, Investigation, Validation, Writing - original draft preparation; **Yuncheng He**: Funding acquisiton, Project administration, Supervision, Writing – review & editing; **Pakwai Chan**: Resources; **Aiming Liu**: Resources; **Qijun Gao**: Resources.

**Acknowledgments**

The research has been supported by National Natural Science Foundation of China (Grant No.: 52178465), Natural Science Foundation of Guangdong Province, China (Grant No.: 2023B1515020117), and the 111 Project of China (Grant No.: D21021).
Conflict of Interest

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
