# Peer review of "Super Typhoons *Mangkhut* (2018) and *Saola* (2023) during landfall: comparison and insights for wind engineering practice"

_EGUsphere, 2024_

## Referee Comment (RC1)

**Review of "Super Typhoons Mangkhut (2018) and Saola (2023) during landfall: comparison and insights for wind engineering practice" by Liu et al.**

This paper is an analysis of wind characteristics of two super typhoons Mangkhut and Saola, aiming to draw relevant insights for wind engineering. This paper is motivated by the absence of super typhoons characteristics in current engineering models, as these models are primarily based on common typhoons. The motivation is good, but the authors fail to address the question posed in the introduction. The emphasis of the paper is on the wind profiles and the gradient winds. The authors claim that low-level jets (LLJs) are identified, but LLJs are a well-documented and common feature in typhoons. The authors suggest that the standardization model can be used to calculate the wind speed of super typhoons, but this merely validates the performance of existing engineering models rather than provide new insights. The paper is also not well written. The lack of key details/definitions and enormous typos make it difficult for the reviewer to follow the study.

The reviewer cannot recommend publication in its current form, as it needs substantial revisions. The authors have to think carefully what they can offer, e.g., which aspects of super typhoons are absent in current engineering practice. Below some suggestions are listed that may help improve the manuscript.

Major comments:
1. There are lots of studies on wind characteristics of these two super typhoons. The authors should cite them.
2. Line 161: Why the Dvorak technique is mentioned but not used in the manuscript? Why no details of the DL method are presented? Why Figure 4 is needed as intensity is already provided in Figure 1? I suggest to remove Figure 4 as it does not provide extra information. Are there any other variables can be provided using the authors' DL method?
3. What do the yellow and blue areas in Figure 6 indicate for?
4. What is $\Delta p_r / \Delta p_c$ in Figure 7? The authors should define these two variables in the main text.
5. Lines 242-243 & 248-249: The authors should cite references to support these lines.
6. Line 252: this paragraph is on Mangkhut, instead of Saola, and there is no second landfall of Mangkhut.
7. Figure 9a needs a colorbar. Why is U-r panel in Figure 9a different from 9b? Can authors provide a U-r panel with observed data for Figure 9a?
8. The three portions of Figure 12 are divided somehow arbitrarily. Can authors provide reasons for this choice?
9. Lines 341-343: A substantial body of literature has already documented the presence of LLJs in typhoons. The authors should cite some of them.
10. Can authors provide a detailed derivation for Eq. (2).
11. Line 360: can authors provide details of the standardization method?
12. Eq. (2): friction due to turbulence should be considered when calculating gradient winds in boundary layers.
13. Line 377: the large gradient winds should be related to pressure gradients instead of LLJ.
14. "modelled wind" is not accurate in section 4.2 as the standardization method is also essentially modelled wind.

15. Lines 376-397: What is super-gradient wind, how does this relate to LLJ, why is gradient wind suddenly related to vertical velocity? The reviewer does not see the logic in these lines.
16. There are a fair number of typos in the manuscript. Suggest the authors have an editor go over the paper. I list some of them below.

Technical comments:

1. Line 29: The first letter of "the" should be capitalized.
2. Line 36: "there conditions" must be grammatically wrong.
3. Line 40: The first letter of "research" should be capitalized.
4. Line 91: What do the authors mean by "Saola got much closer to the Southeastern coastline of China than Mangkhut before making landfall,"? Is the track of Saola closer?
5. Lines 157-159: there must be some typos, given the authors said Saola has an eye but no eye is observed.

---

## Author Comment (AC1)

**In reference to "Super Typhoons Mangkhut (2018) and Saola (2023) during landfall: comparison and insights for wind engineering practice":**

The authors received the review comments from one anonymous reviewer and the editorial board on Dec 3, 2024. We are grateful to the reviewers and the editorial staff for your valuable comments on this manuscript. We will respond to each of them and the relevant corrections are listed below. The relevant corrections are highlighted in red in the second submission of the manuscript, and the responses to the detailed comments are provided below.

**General comments:**

**This paper is an analysis of wind characteristics of two super typhoons Mangkhut and Saola, aiming to draw relevant insights for wind engineering. This paper is motivated by the absence of super typhoons characteristics in current engineering models, as these models are primarily based on common typhoons. The motivation is good, but the authors fail to address the question posed in the introduction. The emphasis of the paper is on the wind profiles and the gradient winds. The authors claim that low-level jets (LLJs) are identified, but LLJs are a well-documented and common feature in typhoons. The authors suggest that the standardization model can be used to calculate the wind speed of super typhoons, but this merely validates the performance of existing engineering models rather than provide new insights. The paper is also not well written. The lack of key details/definitions and enormous typos make it difficult for the reviewer to follow the study.**

**The reviewer cannot recommend publication in its current form, as it needs substantial revisions. The authors have to think carefully what they can offer, e.g., which aspects of super typhoons are absent in current engineering practice. Below some suggestions are listed that may help improve the manuscript.**

Response:

Thanks for your fair and objective evaluation. We highly value your feedback and acknowledge the shortcomings in the current manuscript. In response to your comments, we have made improvements to the manuscript as follows:

1. We have optimized the introduction section of the manuscript. This study aims to compare two typhoons, discussing the differences in their horizontal structures, wind profiles, gradient winds, low-level jets, and other characteristics from an engineering perspective. Our goal is to provide a reference for assessing typhoon risks within the engineering community. The revised introduction is as follows:

"In the design of anti-TC phase for wind turbines, extreme wind velocities are a critical consideration (Liu et al., 2019; Sheng et al., 2021). Guided by design principles centered around extreme wind speeds, researchers have extensively investigated the environmental loads and dynamic responses of turbines under extreme conditions (Gong et al., 2024; Li et al., 2019; Chen et al., 2020). These findings have significantly informed the design of TC-resistant turbines in coastal and offshore areas. The design process typically begins with an initial estimation of TC risk, defined by design wind speeds corresponding to various return periods, which is then used to assess wind loads and wind-induced responses for the ultimate limit states of the turbine (e.g., I.E.C., 2019).

"However, alongside focusing on extreme wind velocities, gaining a deep understanding of the internal structure and evolution of typhoons is equally important for enhancing the safety of wind turbines. As a complex weather system, typhoons contain a wealth of meteorological elements such as temperature, humidity, pressure fields, and vertical and horizontal wind speed components. These elements interact with each other, collectively determining the intensity and path changes of the typhoon. For instance, the axial-asymmetry of TC pressure field can significantly influence local wind speed distributions, adding complexity to the prediction of typhoon behavior (He et al., 2020). Due to the high variability exhibited by typhoons during their spatiotemporal evolution, accurately assessing their impact on wind turbines poses a significant challenge (Ren et al., 2022). Therefore, evaluating the specific effects of individual typhoons on wind turbines becomes particularly crucial during the operational maintenance phase."(Line 39)

2 As the reviewer pointed out, low-level jets (LLJs) are indeed a common characteristic of typhoons. However, typhoon models commonly employed in the engineering field often neglect the evaluation errors in gradient wind assessments that can result from LLJs. Consequently, we hope to explore and discuss these evaluation errors through the analysis of two super typhoons.

The discussion of standardized models in the manuscript is not solely focused on validating their performance. In fact, these standardized models exhibit noticeable deviations under high wind speed conditions. The purpose of discussing the standardized model in the manuscript is to highlight the differences in gradient winds between super typhoons and monsoons, as well as among different super typhoons. Additionally, it aims to explore potential errors and their causes when traditional engineering models are used to estimate the gradient winds of super typhoons. To address these points, we have added supplementary explanations in the relevant sections of the manuscript as follows:

"Thus from the engineering standpoint, one should take great cares when assuming unchanged gradient wind and converting between TC surface winds with varied terrain setups in a way similar to the one for monsoons."(Line 401)

3 In response to your comments on writing quality and details, we will carefully proofread the entire manuscript to correct typos and inaccurately defined technical terms, and address these issues point by point in the following sections.

Finally, we appreciate the opportunity you have given us to improve this work. We commit to making comprehensive revisions based on your suggestions, striving to ensure that this paper provides valuable contributions to the field of wind engineering.

**Major Comments:**

**1. There are lots of studies on wind characteristics of these two super typhoons. The authors should cite them.**

Response:

We would like to thank the reviewer for this suggestion. In the revised manuscript, we have added several references pertaining to both typhoons, aiming to provide readers with a more comprehensive understanding of the details surrounding these events. The modified section is as follows:

"Super Typhoon *Mangkhut* in 2018 ( Zheng et al., 2024) and Super Typhoon *Saola* in 2023 (Lo et al., 2024) are the two strongest TC events that have attacked South China during the past

years."(Line 57)

"…its peak intensity prior to making landfall on Luzon on 14th, with the maximum sustained surface wind estimated at 70 m/s (He et al., 2020)."(Line 73)

"…the storm traversed Hong Kong with the central maximum winds reaching 58 m/s, and prompting the issuance of the highest-level typhoon red warning signal by the China Meteorological Administration (CMA) (Li et al., 2024)." (Line 83)

**2. Line 161: Why the Dvorak technique is mentioned but not used in the manuscript? Why no details of the DL method are presented? Why Figure 4 is needed as intensity is already provided in Figure 1? I suggest to remove Figure 4 as it does not provide extra information. Are there any other variables can be provided using the authors' DL method?**
Response:

Thanks for reviewer's careful review and valuable comments, we agree that it contains similar information to what is presented in Figure 2. In fact, the DL method introduced in the manuscript does not provide additional variables. Taking this into consideration, we have decided to remove the sections concerning the Dvorak technique as well as the DL method from the updated manuscript.

**3. What do the yellow and blue areas in Figure 6 indicate for?**
Response:

We apologize for any confusion caused in previous submission. In the revised Figure 5 (previously Figure 6), the yellow and blue areas represent two distinct states within the secondary circulation of tropical cyclones: the yellow area indicates measurement points located in the descending air current zone, characterized by high temperatures and low humidity; whereas the blue area corresponds to the ascending air current zone, which is marked by relatively lower temperatures and higher humidity. By employing these two colors to differentiate between the states, we aim to assist readers in gaining a more intuitive understanding of the secondary circulation structure of tropical cyclones. This color coding also facilitates comparative analysis between the broad cloud system of Typhoon Mangkhut and the clear spiral rainbands of Typhoon Sula.

In the updated manuscript, we have added relevant explanations to avoid any misunderstanding, as detailed below:

"The hot and dry subsidence flows (Yellow area in Figure 5) became most evident when the radial distance was about 750 km, with the maximum change of $T_{air}$ and $RH$ equal to 8°C and -30%; while they got minimized when the distance was 300 km, after which the study sites were influenced by stratified rainbands and the atmosphere turned to be saturated (Blue area)." (Line 193)

**4. What is $\Delta p/\Delta p$ in Figure 7? The authors should define these two variables in the main text.**
Response:

Thanks for the reviewer's suggestions. The intention of this figure is to illustrate the distribution of typhoon pressure along the radial distance, which aids in understanding and distinguishing the wind fields of the two typhoons. The label in the original figure was a typo. In the corrected manuscript, we have revised the labels on the image accordingly, as Figure 1 (Figure 6 in the updated manuscript). The variables defined in the updated manuscript as follow:

"where $P(r)$ denotes the mean-sea-level pressure at radial distance $r$, $P_0$ denotes the pressure at the TC center, ambient pressure; $\Delta P_0$ is the difference between the ambient pressure and $P_0$ (or the central pressure deficit), $B$ is a non-dimensional coefficient which governs the shape

of the radial profile." (Line 211)

[Figure]

Figure 1: Modeling results of TC pressure field for *Mangkhut* and *Saola*: (a) normalized radial profiles, (b) RMW and B (0-h marks the time when the storm just got landfall)

**5. Lines 242-243 & 248-249: The authors should cite references to support these lines.**

Response:

We would like to thank the reviewer for this suggestion. In the revised manuscript, we have added new references to this paragraph as follows:

"It has been well acknowledged that for intense TCs over deep seawater, an outer eyewall may form outside the initial (or inner) eyewall, and the storm demonstrates a concentric eyewall structure (Houze et al., 2007)." (Line 236)

"Meanwhile, the outer eyewall tends to shrink and gradually replace the inner eyewall. The above process is termed as the eyewall replacement (ER), which has been observed in the evolution of many super typhoons (Wang et al., 2024; Ling et al., 2024)."(Line 240)

**6. Line 252: this paragraph is on Mangkhut, instead of Saola, and there is no second landfall of Mangkhut.**

Response:

Thanks for reviewer`s correction. The paragraph was intended to describe Typhoon Mangkhut, and we acknowledge that this was a typo. In the updated manuscript, we have corrected this typo in the updated manuscript as follow:

"However, the storm failed to finish the ER cycle, as: (a) it first made landfall on Luzon, which destroyed its inner structure (He et al., 2020), and (b) after the first landfall, *Mangkhut* moved to the South China Sea and approached to the southeast coastline of China where the ambient conditions became unfavorable for its further development." (Line 246)

If I have understood correctly, the reviewer believes that Typhoon Mangkhut did not have a second landfall. To the best of my knowledge, Typhoon Mangkhut made its initial landfall along the coast of China's South China Sea. Typhoon Mangkhut had already made its first landfall on Luzon Island in the Philippines, as shown in Figure 2. This process has been extensively documented in various studies and official meteorological records:

[Figure]

Fig.2. Best track of Typhoon Mangkhut (He et al., 2020)

He, J.Y., He, Y.C., Li, Q.S., et al., 2020. Observational study of wind characteristics, wind speed and turbulence profiles during Super Typhoon Mangkhut. Journal of Wind Engineering and Industrial Aerodynamics. 206: 105362

Cao, M., Wang, Q.Y., 2022. Observed near-inertial wares and shears east of Luzon during Typhoon Mangkhut and Yutu in 2018. Deep-sea Research Part II. 205: 105185.

**7. Figure 9a needs a colorbar. Why is U-r panel in Figure 9a different from 9b? Can authors provide a U-r panel with observed data for Figure 9a?**
Response:

Thank for reviewer`s suggestions. The data for Figure 9 are sourced from the Hong Kong Observatory, with Figure 9(a) originating from 2018 and Figure 9(b) from 2023. Advances in technology have resulted in differences in the visual presentation of the two images. Currently, specific information about Typhoon Mangkhut is quite limited, and due to the unavailability of precise numerical data, we are unable to add a colorbar or U-r panel to Figure 9(a).

According to the above reasons, the authors have conducted a simple analysis Based on the available information, and provided an illustration of the observable dual eyewall structure and partial eyewall replacement cycle of Typhoon Mangkhut in the figure, ensuring that readers can also identify the structural changes of Typhoon Mangkhut.

**8. The three portions of Figure 12 are divided somehow arbitrarily. Can authors provide reasons for this choice?**
Response:

We appreciate the meticulous review from the reviewer. In this section, we have divided the two typhoons into three radial zones based on the vertical wind velocity: the outer portion, the middle portion and inner portion, as shown in Figure 3. in the outer portion, the vertical wind velocity is greater than zero, indicating no observed rainfall; in the middle portion, the vertical wind velocity ranges from 0 to -5 m/s, which typically points to intermittent rainfall activity; while in the inner portion, the vertical wind velocity lies between -5 and -10 m/s, marking continuous strong precipitation events. As the reviewer pointed out, such zonal division carries a degree of subjectivity, so the specific boundaries of these zones should be considered approximate. To clarify this point and avoid potential confusion, we have supplemented and refined the relevant descriptions in the revised manuscript:

"The radial scope with respect to the TC center at the study site below 5 km can be basically divided into 3 portions according to *W*: the inner portion dominated by torrential rains (about -280<*r*<180 km for *Mangkhut* and -160<*r*<80 km for *Saola*), the outer portion with light or no rains (outside about -500<*r*<470 km for *Mangkhut* and outside about -250<*r*<250 km for *Saola*; corresponding to the TC periphery that was governed by convective cloud cells)…"(Line304)

"It is important to note that while this method of zonation carries a degree of subjectivity, it remains representative for understanding the horizontal structure of the two typhoons."(Line 310)

[Figure]

**Fig.3. Vertical profiles of nominal vertical speed (*W*) and associated signal-to-noise ratio (*SNR*) at SSP—(a): *W* for *Mangkhut*, (b): *W* for *Saola***

**9. Lines 341-343: A substantial body of literature has already documented the presence of LLJs in typhoons. The authors should cite some of them.**
Response:

Thanks for the reviewer's suggestions. We have added relevant references to the revised manuscript. The modified manuscript is as follows:

"For both typhoons, the vertical profiles demonstrate a low-level-jet (LLJ) structure. LLJ is a band of strong winds frequently observed in tropical cyclones, typically occurring in the near-surface layer (Li et al., 2019; Hao et al., 2024). This feature differs significantly from the traditional depiction of wind profiles, , which assumes unchanged winds above the gradient height." (Line 341)

**10. Can authors provide a detailed derivation for Eq. (2).**
Response:

Thanks for the reviewer's suggestions. We have re-edited the relevant content and added the derivation process of the gradient wind in the updated manuscript as follows:

$$V_g = V_{cf} + \sqrt{V_{cf}^2 + \frac{r}{\rho}\frac{\partial p}{\partial r}} \tag{2}$$

$$V_{cf} = \frac{1}{2}(U_T \sin(\alpha) - fr) \tag{3}$$

$$\frac{\partial p}{\partial r} = \frac{\Delta P_0 B}{r}\exp\left[-\left(\frac{RMW}{r}\right)^B\right]\left(\frac{RMW}{r}\right)^B \tag{4}$$

where, $V_g$ signifies the gradient wind speed with respect to the TC direction offset angle $\alpha$

and the radial distance $r$, $U_{\mathrm{T}}$ denotes the TC translational speed, $f$ represents the Coriolis coefficient ($f = 2\Omega \sin(\phi)$, $\Omega = \pi/(12 \times 3600)$ symbolizes the Earth angular velocity of rotation, $\varphi$ denotes latitude) and $\rho$ indicates air density. (Line 366)

**11:Line 360: can authors provide details of the standardization method?**

Response:

Thank you for your comments. The standardization method discussed in this paper is based on extensive field wind data collected from over 50 meteorological stations in Hong Kong. These data primarily include measurements from Light Detection and Ranging (LiDAR) and surface anemometers. We established a mapping relationship between near-surface wind speeds and actual gradient winds under different wind directions, presented in the form of correction factors. Through this method, we can accurately derive local gradient wind speeds from surface wind speed and direction data, even under complex terrain conditions.

**12:Eq. (2): friction due to turbulence should be considered when calculating gradient winds in boundary layers.**

Response:

Thank you for your insightful comments. In our study, we focus on the idealized scenario of gradient winds in the free atmosphere, where friction effects are typically negligible. Our primary objective is to analyze and compare the large-scale features of super typhoons, such as horizontal structures, wind profiles, and LLJs, without the complicating factors introduced by boundary layer processes. Therefore, we have chosen to use the classical gradient wind approximation, which assumes a balance between pressure gradient force and Coriolis force, without incorporating frictional effects.

We acknowledge that including friction due to turbulence is crucial for detailed modeling of near-surface wind conditions, especially for engineering applications like wind turbine design. However, this level of detail is beyond the scope of our current analysis, which aims to provide a broader understanding of typhoon dynamics.

Nonetheless, we appreciate the importance of considering these factors in certain contexts and have added a note in the manuscript to discuss the limitations of our approach and highlight areas for future research. Specifically, we mention:

"…It is important to note that, since this study does not focus on near-surface conditions, the formula does not consider frictional effects due to turbulence." (Line 372)

**13:Line 377: the large gradient winds should be related to pressure gradients instead of LLJ.**

Response:

We fully agree with the reviewer's point that large gradient winds are primarily caused by pressure gradients in the horizontal direction. In this section of the manuscript, we intended to convey that super-gradient winds are widely present in the eyewall and rainband regions of tropical cyclones and manifest as LLJ (low-level jets) in the vertical profile of the storm.

We have revised the corresponding description in the manuscript to avoid any misunderstanding as follows:

"Note that super-gradient wind exists widely in the eyewall and rainband regions of TCs, which

manifest as LLJ in TC vertical wind profile (Kepert, 2010)." (Line 379)

**14: "modelled wind" is not accurate in section 4.2 as the standardization method is also essentially modelled wind.**
Response:

Thank you for pointing out the potential ambiguity in Section 4.2. We agree that the term "modelled wind" can be misleading in this context, especially given that the standardization method also produces a form of modelled wind. So we have revised the text to distinguish between "standardized" which refers to the wind speeds derived from the standardization method, and "calculated" which refers to outputs from the gradient wind calculation formula. This distinction helps to avoid confusion and clearly delineates the different approaches used in our analysis. The updated text and figure now as follows:

"As illustrated in Figure 13(a), the calculated results agree better with the measured gradient winds for *Saola* than for *Mangkhut*. Note that super-gradient wind exists widely in the eyewall and rainband regions of TCs, …" (Line 379)

"This phenomenon partially accounts for the discrepancies between the calculated results and measurements shown in Figure 13(a)." (Line 381)

"However, the measured results for *Mangkhut* are too large compared to the calculated values, and considerable errors should exist in the calculated gradient wind." (Line 388)

[Figure]

**Figure 13: TC gradient wind measured by DRWPs compared with calculated results (a) and those deduced via the standardization method (b).**

**15. Lines 376-397: What is super-gradient wind, how does this relate to LLJ, why is gradient wind suddenly related to vertical velocity? The reviewer does not see the logic in these lines.**
Response:

We apologize for any confusion caused in original manuscript. Below, we provide a detailed explanation of the concepts related to super-gradient winds, LLJ, and the vertical wind component (W) to clarify these relationships.

Super-gradient winds refer to situations where actual wind speeds exceed those predicted by geostrophic balance theory. This phenomenon commonly occurs in the eyewall and rainband regions of tropical cyclones. The occurrence is driven by inertial centrifugal force exceeding the combined effect of Coriolis and pressure gradient forces, leading to significantly increased horizontal wind speeds. In these regions, LLJ (low-level jet) represents a specific manifestation of super-gradient winds, characterized by intense wind bands in the lower atmosphere with velocities much higher than surrounding areas.

The traditional gradient wind equations primarily focus on horizontal wind speed balance, while the vertical wind component (W) plays a crucial role in tropical cyclones, especially within the eyewall region. Strong updrafts in these areas significantly enhance local pressure gradients, causing actual wind speeds to surpass geostrophic wind speeds, thereby forming super-gradient winds.

To improve clarity for readers, we have revised the relevant section as follows:

"This phenomenon partially accounts for the discrepancies between the calculated results and measurements shown in Figure 13(a). Super-gradient winds are characterized by actual wind speeds that exceed those predicted by geostrophic balance theory, typically observed in the eyewall and rainband regions of tropical cyclones (Kepert, 2010). In this context, LLJ (low-level jet) manifests as an intense wind band in the lower atmosphere. Notably, the vertical wind component (W) plays a critical role in tropical cyclones by influencing thermal structure, momentum transport, and inertial centrifugal force, indirectly promoting the formation of super-gradient winds. Therefore, Equation (2), which does not account for the vertical wind component (W), may contribute to the model's inability to fully reproduce the observed characteristics of super-gradient winds."(Line 381)

**16. There are a fair number of typos in the manuscript. Suggest the authors have an editor go over the paper. I list some of them below.**

Response:

Thanks for the reviewer`s carefully examination. In the update manuscript, the authors have reviewed the entire text and corrected the typos. Below are the corrections for these typos along with the revised manuscript:

**1. Line 29: The first letter of "the" should be capitalized.**

"The impact of TCs is especially noteworthy (Matsui et al., 2002)." (Line 29)

**2. Line 36: "there conditions" must be grammatically wrong.**

"These conditions also impose heightened requirements on the TC-resilience capabilities of offshore wind farms during both their construction and operational maintenance phases" (Line 36)

**3. Line 40: The first letter of "research" should be capitalized.**

"Guided by design principles centered around extreme wind speeds, researchers have extensively investigated the environmental loads and dynamic responses of turbines under extreme conditions (Gong et al., 2024; Li et al., 2019; Chen et al., 2020)." (Line 40)

**4. Line 91: What do the authors mean by "Saola got much closer to the Southeastern coastline of China than Mangkhut before making landfall,"? Is the track of Saola closer?**

"(2) Unlike *Mangkhut*, the path of *Saola* stayed closer to the southeastern coastline of China before finally making landfall. with the nearest distance between HK and the TC's center track being ~30 km for *Saola* and ~100 km for *Mangkhut*;" (Line 96)

**5. Lines 157-159: there must be some typos, given the authors said Saola has an eye but no eye is observed.**

"By contrast, no such typical TC eye and convective tower were observed for *Mangkhut*."(Line 163).

---

## Author Comment (AC2)

**In reference to "Super Typhoons Mangkhut (2018) and Saola (2023) during landfall: comparison and insights for wind engineering practice":**

The authors received the review comments from one anonymous reviewer and the editorial board on Feb 13, 2025. We are grateful to the reviewers and the editorial staff for your valuable comments on this manuscript. We will respond to each of them and the relevant corrections are listed below. The relevant corrections have been highlighted in red in the revised manuscript, and the responses to the detailed comments are provided below.

**General comments:**

**This manuscript offers a comprehensive comparison of the wind field characteristics of two super typhoons, Mangkhut and Hato, aiming to analyze their differences from a wind engineering perspective, including horizontal and vertical wind field structures. It also describes features such as dual eyewalls, gradient winds, and low-level jets for both typhoons. Overall, this is a well-written article, but there are several areas that require modification.**

**Response:**

Thanks for your fair and objective evaluation. We highly value your feedback and acknowledge the areas for improvement in the current manuscript. In response to your comments, we have made improvements to the manuscript as follows:

**1 The introduction does not clearly articulate the purpose of this study. Given that the authors are not aiming to develop a more accurate typhoon engineering model, it is suggested that the manuscript's introduction be revised to better reflect the study's objectives.**

**Response:**

We appreciate the reviewer's suggestions. The primary objective of this study is to compare two typhoons with the aim of identifying differences in their wind fields from an engineering perspective. This research seeks to provide valuable references for typhoon-resistant design, particularly focusing on offshore wind turbines. In light of the aforementioned suggestions, we have revised the introduction to better reflect our research objectives as follows:

"However, alongside focusing on extreme wind velocities, gaining a deep understanding of the internal structure and evolution of typhoons is equally important for enhancing the safety of wind turbines. As a complex weather system, typhoons contain a wealth of meteorological elements such as temperature, humidity, pressure fields, and vertical and horizontal wind speed components. These elements interact with each other, collectively determining the intensity and path changes of the typhoon. For instance, the axial-asymmetry of TC pressure field can significantly influence local wind speed distributions, adding complexity to the prediction of typhoon behavior (He et al., 2020). Due to the high variability exhibited by typhoons during their spatiotemporal evolution, accurately assessing their impact on wind turbines poses a significant challenge (Ren et al., 2022). Therefore, evaluating the specific effects of individual typhoons on wind turbines becomes particularly crucial

during the operational maintenance phase." (Line 39)

**2 The manuscript mentions work related to artificial intelligence; however, it is unclear how these preliminary studies contribute to the current research.**

**Response:**
We concur with the reviewer's assessment that the role of AI in this study is minimal. Consequently, we have decided to remove the relevant sections from our manuscript. The revised portions of our manuscript are now presented as follows:

"…Instead, the central area before landfall was occupied by an irregular region of loose or patchy cloud which carried some characteristics of TC eye.

To detail the main-circulation (in particular the rain-band) structure of the two typhoons beneath the cloud shield, Figure 4 exhibits the echo-grams from the ground-based weather radar…" (Line 164)

**3 Line 241, it is recommended that the discussion of eyewall replacement cycles be supported by additional references.**

**Response:**
We appreciate the reviewer's suggestions. We have completely rewritten this section, incorporating additional information regarding eyewall replacement cycles, and have added several references for further context and support. The revised manuscript is now presented as follows:

"It has been well acknowledged that for intense TCs over deep seawater, an outer eyewall may form outside the initial (or inner) eyewall, and the storm demonstrates a concentric eyewall structure (Houze et al., 2007)." (Line 236)

"Meanwhile, the outer eyewall tends to shrink and gradually replace the inner eyewall. The above process is termed as the eyewall replacement (ER), which has been observed in the evolution of many super typhoons (Wang et al., 2024; Ling et al., 2024)."(Line 240)

**4 Line 370, the authors emphasize the role of super-gradient winds in enhancing low-level jets. However, the manuscript lacks a detailed explanation of how super-gradient winds specifically influence the low-level jet characteristics within typhoons. The reviewer suggests further elaboration on this topic.**

**Response:**
We appreciate the reviewer's constructive suggestions. In fact, the characteristics of the low-level jet in the vertical wind profile are not directly caused by supergradient winds. However, supergradient winds do lead to increased momentum transport and enhanced turbulence within the typhoon, thereby further intensifying the strength of the low-level jet. In the revised manuscript, we provide a more detailed explanation as follows:
While the low-level jet within a typhoon is not directly induced by supergradient winds, the latter

significantly enhances local momentum transport and increases turbulent mixing within the wind field, leading to a further enhancement of the low-level jet (Line 383)

**5 Regarding the impact of super-gradient winds on gradient winds within the typhoon wind field, can the authors locate literature that quantifies this effect?**

**Response:**

We appreciate the reviewer's suggestions. In response to the impact of super-gradient winds on gradient winds, we have reviewed existing research findings. According to Kepert's study, super-gradient winds can influence gradient winds by more than 10%. Based on this conclusion, we have supplemented and elaborated on the relevant sections of our manuscript as follows:

"The investigative outcomes presented by Kepert (2001b) suggest that this disparity might potentially exceed a magnitude of ten percent." (Line 378)

**6 There are issues with some figures, such as Fig. 7(a), where the parameter represented by the y-axis is not labeled; the same issue appears in Fig. 14.**

**Response:**

We appreciate the reviewer's meticulous examination. If we understand correctly, the reviewer points out that the axes in these figures have not been adequately explained in the manuscript. In response to this issue concerning the two figures, we have made the following revisions:

For Fig. 7(a),We have modified the picture to ensure that the label of the picture conforms to the relevant description in the manuscript:

[Figure]

Figure 6: Calculated results of TC pressure field for *Mangkhut* and *Saola*: (a) normalized radial profiles, (b) RMW and B (0-h marks the time when the storm just got landfall)

For Fig. 14, we have provided additional explanations in the paper as follows:

as shown in Figure 13(a), Where, $U_{max}$ represents the measured maximum wind speed, which denotes the gradient wind speed. The measured gradient winds are also compared with their counterparts obtained via the standardization method developed by He et al. (2014), as shown in

Figure 13(b), Where $U_{\text{stand}}$ represents the normalized gradient wind speed and $U_{\text{DRWP}}$ denotes the measured results.

**7 The manuscript contains typographical errors (e.g., Line 252 should refer to Typhoon Mangkhut rather than Typhoon Saola) and inconsistencies in font size, such as Lines 215-220 and 365-368.**

**Response:**

We are grateful for the reviewer's meticulous examination. We have corrected the typos identified in the manuscript as follows:

**"…** and (b) after the first landfall, *Mangkhut* moved to the South China Sea and approached to the southeast coastline of China where the ambient conditions became unfavorable for its further development" (Line 252)